# miR2118-dependent U-rich phasiRNA production in rice anther wall development

Saori Araki[1,9], Ngoc Tu Le [2,9], Koji Koizumi[3,9], Alejandro Villar-Briones[4], Ken-Ichi Nonomura[5,6], Masaki Endo[7], Haruhiko Inoue[7,8], Hidetoshi Saze[2] & Reina Komiya [1,8✉]

Reproduction-specific small RNAs are vital regulators of germline development in animals and plants. MicroRNA2118 (miR2118) is conserved in plants and induces the production of phased small interfering RNAs (phasiRNAs). To reveal the biological functions of miR2118, we describe here rice mutants with large deletions of the miR2118 cluster. Our results demonstrate that the loss of miR2118 causes severe male and female sterility in rice, associated with marked morphological and developmental abnormalities in somatic anther wall cells. Small RNA profiling reveals that miR2118-dependent 21-nucleotide (nt) phasiRNAs in the anther wall are U-rich, distinct from the phasiRNAs in germ cells. Furthermore, the miR2118-dependent biogenesis of 21-nt phasiRNAs may involve the Argonaute proteins OsAGO1b/OsAGO1d, which are abundant in anther wall cell layers. Our study highlights the site-specific differences of phasiRNAs between somatic anther wall and germ cells, and demonstrates the significance of miR2118/U-phasiRNA functions in anther wall development and rice reproduction.

[1] Science and Technology Group, Okinawa Institute of Science and Technology Graduate University (OIST), 1919-1 Tancha, Onna-son, Okinawa 904-0495, Japan. [2] Plant Epigenetics Unit, OIST, 1919-1 Tancha, Onna-son, Okinawa 904-0495, Japan. [3] Imaging Section, OIST, 1919-1 Tancha, Onna-son, Okinawa 904-0495, Japan. [4] Instrumental Analysis Section, OIST, 1919-1 Tancha, Onna-son, Okinawa 904-0495, Japan. [5] Plant Cytogenetics, Department of Gene Function and Phenomics, National Institute of Genetics, 1111 Yata, Mishima, Shizuoka 411-8540, Japan. [6] Department of Life Science, Graduate University for Advanced Studies/Sokendai, 1111 Yata, Mishima, Shizuoka 411-8540, Japan. [7] Institute of Agrobiological Sciences, National Agriculture and Food Research Organization, 1-2 Owashi, Tsukuba, Ibaraki 305-8634, Japan. [8] Japan Science and Technology Agency, PRESTO, 4-1-8 Honcho, Kawaguchi, Saitama 332-0012, Japan. [9]These authors contributed equally: Saori Araki, Ngoc Tu Le, Koji Koizumi. ✉email: reina.komiya@oist.jp

Small RNA is involved in various developmental processes as well as in genome defense in animals and plants[1,2]. Based on the source and function, these small RNAs are classified as either microRNA (miRNA), repeat-associated small interfering RNA (rasiRNA), or phased small interfering RNA (phasiRNA) in plants[3]. Functionally, the small RNAs are loaded into Argonaute (AGO) proteins to form the RNA-induced silencing complex, which can induce RNA silencing[1,4]. miRNAs generally regulate the translation of complementary messenger RNA (mRNA)[5]. Nevertheless, some miRNA species induce the generation of secondary phasiRNAs, also known as *trans*-acting siRNA (tasiRNAs), in *Arabidopsis*[6,7]. In phasiRNA/tasiRNA biogenesis, a 22-nucleotide (nt) miRNA induces the cleavage of phasiRNA precursor-RNAs (*PHASs*). The cleavage of *PHAS* initiates secondary phasiRNA production by the synthesis and processing of double-stranded RNAs at 21- or 24-nt intervals by dicer-like (DCL) proteins[8–12].

miR2118 is a 22-nt miRNA that is widely conserved in angiosperms and gymnosperms, and is involved in the production of 21-nt phasiRNAs. In monocots, miR2118 initiates the production of 21-nt phasiRNAs, especially in reproductive tissues[8]. In rice, the 21-nt phasiRNAs are derived from over 700-long intergenic non-coding RNAs (lincRNAs)/*21PHASs* that are abundant in young anthers from primordial germ-cell initiation to meiosis[10]. On the other hand, when meiosis is initiated, 24-nt phasiRNAs are produced by miR2275 induction and DCL3b/5 processing. The spatio-temporal production of 21-nt pre-meiotic and 24-nt meiotic phasiRNAs has been reported during maize reproduction[13]. Interestingly, even though the phasiRNA biogenesis triggered by miR2118 cleavage is conserved in both monocots and dicots, miR2118 targets reproductive lincRNAs in monocots, while it generally targets protein-coding RNAs, especially derived from disease-resistant genes in dicots[14–16]. Recently, the tomato miR482/2118 family was reported to affect disease resistance via regulation of mRNAs of leucine-rich repeat genes[17]. However, the molecular functions of miR2118 and 21-nt phasiRNAs during reproductive stages, particularly in monocots, are not fully understood.

The anther is a part of the stamen, consisting of the germ (pollens) and soma (anther wall cell layers). Defects of anther development lead to pollen sterility, and therefore, reproductive control via anther development is a major factor in determining plant yield. During early meiosis in rice, microspores are formed in the anther, which is surrounded by the following four somatic anther wall layers: the epidermis, endothecium, middle layer, and tapetum. Factors required for the development of the inner tapetum layer, which plays a key role in supplying nutrients to the microspore via programmed cell death (PCD), have been identified[18–20]. It has been reported that the AGO protein, MEIOSIS ARRESTED AT LEPTOTENE 1 (MEL1), binds to phasiRNAs during reproduction in rice[10,21]. MEL1 mutants exhibit defects in homologous chromosome synapsis in meiocytes, but no clear defects have been observed in anther wall development, thereby indicating that it primarily acts on early meiosis in germ cells[22,23]. Thus, the molecular mechanisms regarding the development of the outer layers of the anther wall and anther wall-specific AGO-small RNAs remain largely unknown.

To understand the function of miR2118 in rice reproduction, we generate rice miR2118 mutants by genome editing. By integrating small RNA profiling, proteome analysis, and 3D-histochemical analyses of miR2118 mutants, we demonstrate the functions of miR2118 in anther wall development, and propose a model for the biogenesis of miR2118-dependent U-phasiRNA with AGO1 subfamilies during male reproduction in rice.

## Results

**Male and female sterility in miR2118 deletion mutants.** In the rice genome, 18 family members of miR2118 are clustered at two regions on chromosome 4 and 11[9], which encode 12 different classes of mature miR2118 (Fig. 1a). To determine the function of miR2118 in rice, we performed gene editing of the miR2118 loci in chromosome 4, with the CRISPR/Cas9 system using the identical sequence of miR2118fjm as a guide RNA (Fig. 1a; for details see "Method" section). We obtained various edited lines with mutations in miR2118 families in the cluster (e.g., Supplementary Fig. 1a). Additionally, we obtained two independent edited lines with a large deletion of 16 kb for the miR2118 cluster, which includes the 14 miR2118 loci between miR2118b and miR2118n with identical sequences to the guide RNA, except for the C/U region at the PAM sequence (Fig. 1a, b; Supplementary Fig. 1b–d). We further obtained two siblings ($T_1$ lines) from one $T_0$ line, which resulted in a total of three deleted lines of miR2118 clusters (Supplementary Fig. 1c). These three lines were identical to the cleavage sites at the miR2118b and miR2118n loci, resulting in a deletion of 16,209 bp of the cluster (Fig. 1b). Whole-genome sequencing of the three lines also confirmed the deletion from miR2118b to miR2118n, while miR2118a at chromosome 4, along with miR2118p, q, and r at chromosome 11, remained intact. Nevertheless, genetic variations among the three lines were found, including single-nucleotide polymorphisms (SNPs) in other genomic regions (Supplementary Fig. 1e; for details see "Method" section). Given the identical deletion of the miR2118 cluster with differences in the genomic background of the three lines, the miR2118b–n deletion lines (hereafter *mir2118*) were named as *mir2118-1-1*, *mir2118-1-2*, and *mir2118-1-3* (*mi-1*, *mi-2*, and *mi-3*, respectively).

*Photoperiodic-sensitive genic male sterility1* (*PMS1T*) is a long non-coding RNA that regulates photoperiodic male sterility via miR2118 cleavage and the production of 21-nt phasiRNAs[24]. The *pms1t* mutant exhibited long-day (LD)-specific sterility in male tissues. The *mir2118* exhibited severe sterility, especially under short-day (SD) conditions (Fig. 1c). We found that *mir2118* anthers were more curled and shorter than wild-type (WT) anthers, and that mature pollens did not contain starch (Fig. 1d–f). In terms of female reproductive tissue, ~18% of *mi-1* and 25% of *mi-2* plants exhibited three or four abnormal stigmas in the pistils (Fig. 1g, h; Supplementary Table 1). Pollination tests of WT ("Nipponbare") and *mir2118* revealed that miR2118 is required for both male and female fertility (Supplementary Table 2). These results suggest that miR2118 is vital for reproductive tissue development in both females and males. Furthermore, the segregation rate of the mutation in the offspring from *mir2118* heterozygous plants exhibited Mendelian inheritance (Supplementary Fig. 2a), and severe sterility was evident in homozygous *mir2118* (Supplementary Fig. 2b), which indicates that the abnormal phenotypes are due to sporophytic effects of *mir2118*.

In the RAP-DB rice genome annotation (https://rapdb.dna.affrc.go.jp/), two hypothetical genes, namely *Os04g0435300* and *Os04g0435475/LOC_04g35550*, are predicted to be inside the miR2118 deleted regions, in addition to four precursor loci of miR2118gf, ih, j, and lk (Supplementary Fig. 3a). However, these transcripts were also expressed in other rice tissues, in contrast to the expression of the miR2118o precursor, which is more specific to the pre-meiotic stages (Supplementary Fig. 3b). This suggests that the reproductive defects in *mir2118* are mainly due to the miR2118b–n deletion, while we could not exclude the possibilities that the loss of *Os04g0435300* and *Os04g0435475/LOC_04g35550* had effects on the observed phenotypes in *mir2118*.

The reproductive phenotypes of *mi-1, mi-2,* and *mi-3* were essentially identical (Supplementary Fig. 2), and no significant

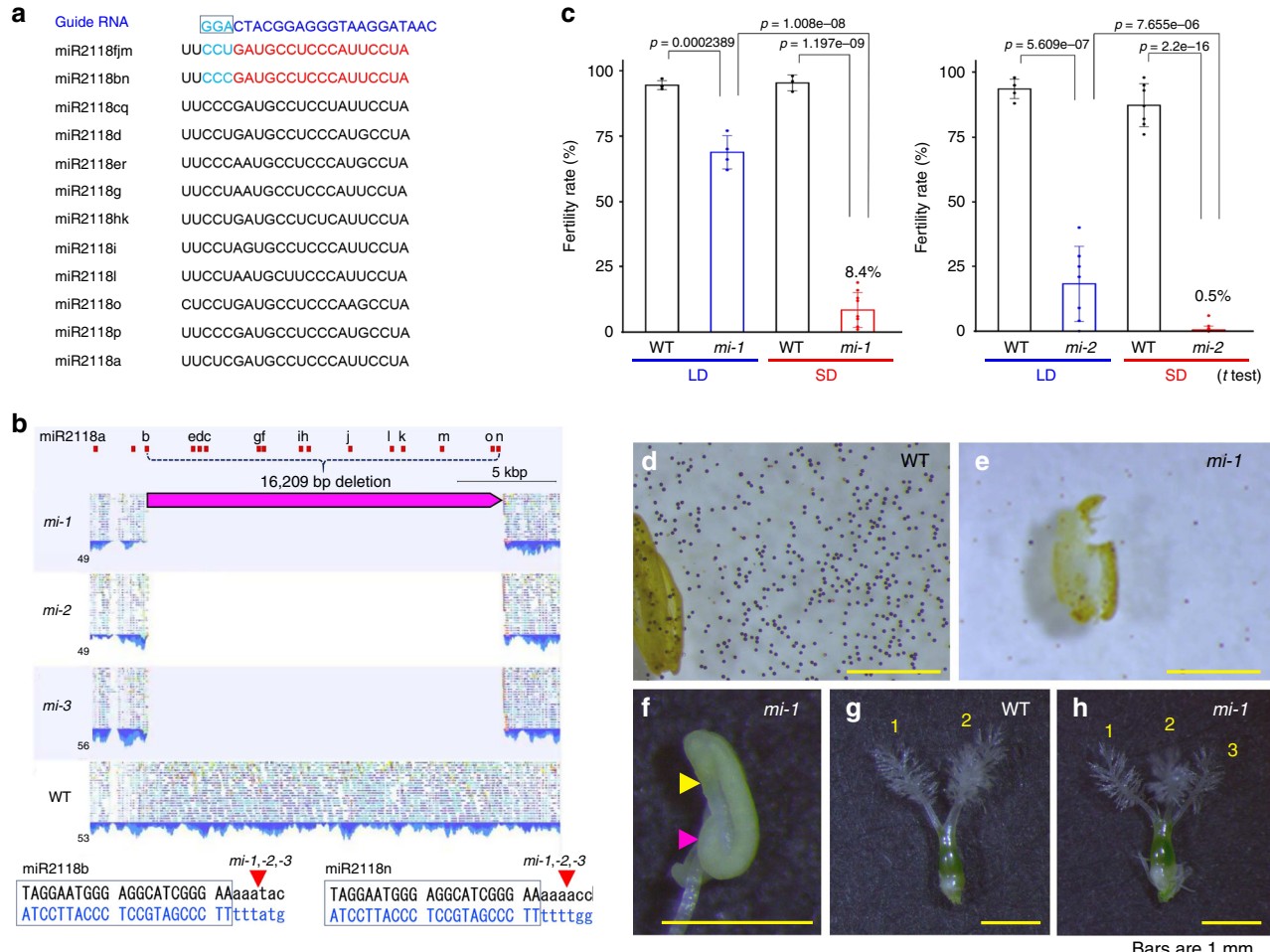

**Fig. 1 miR2118 deletion mutants exhibit male and female sterility. a** Sequence alignment of the 12 mature miR2118 family members in rice, and Cas9-guide RNA used for genome editing. The Cas9-guide RNA complementary sequence (blue) is mostly conserved in miR2118bn and miR2118fjm (red). The PAM sequence is shown in light blue in a black box. **b** Genome sequence read mapping in the miR2118 cluster region on rice chromosome 4 in the *mi-1*, *mi-2*, and *mi-3* lines. In all, 16,209 bp from miR2118b to miR2118n were deleted in the *mir2118* lines. **c** Fertility of Nipponbare (WT) and two *mir2118* mutants grown under long-day (LD) and short-day (SD) conditions. Data are mean ± SD of more than three biological replicates. Student's *t* test. **d, e** Mature pollens stained with iodine-potassium iodide of the WT and *mir2118* line. **f** An anther of *mi-1*. Yellow arrowhead indicates the shorter inner locule. Magenta arrowhead indicates the curled outer locule. **g, h**. Pistils of the WT and *mi-1* line. An additional stigma was observed in the *mir2118* line. Yellow bars indicate 1 mm.

differences were found in plant height between WT and *mi-1* or *mi-2* under SD and LD conditions (Supplementary Fig. 4). Although 90 genetic variants in *mi-1*, 359 in *mi-2*, and *179 in mi-3* were detected compared to the Nipponbare genome (Supplementary Fig. 1e), no mutation was identified within the *21PHAS* loci (body) in the three lines, while six and two variations were detected within 1 kp upstream/downstream of *21PHAS* in *mi-2* and *mi-3*, respectively (Supplementary Fig. 1f). These and other unlinked genetic variants may cause the variable expressivity of the phenotypes among the *mir2118* lines (e.g., Figs. 2–4 and Supplementary Fig. 6c). We focused on the *mi-1* and *mi-2* lines for further analyses.

**Defects of epidermis development in the anther of *mir2118*.**
We further investigated the role of miR2118 during anther development, given the prominent developmental abnormalities in the mutant. The developmental stage of the anther is classified into six stages for the analysis, as shown in Table 1. We found that the anther length of *mir2118* was shorter than that of WT even at Stage 2, which is approximately the early meiotic stages

(Fig. 2a). To investigate the overall structure of anthers, the 0.5 mm anther at Stage 2 was visualized in 3D images using Lightsheet microscopy by detecting autofluorescence over 585 nm (Fig. 2b–i). The anther comprises four anther locules, and the two outer locules are generally longer than the two inner locules in rice. One locule comprises pollen mother cells and four anther wall cell layers (epidermis, endothecium, middle layer, and tapetum). *mir2118* showed malformation of shorter inner anther locules, while the two outer anther locules exhibited curled shapes in the *mir2118* mutants (Fig. 2c–e; Supplementary Movies 1–3). In the longitudinal section of the 3D anther image, the autofluorescence signals of the epidermis and/or endothelium were evident in the WT (Fig. 2f; a white arrow). However, these signals were extremely weak in *mir2118* mutants and narrowly observed (Fig. 2g–i). We further examined the cell size and shape of the epidermis of 0.4 mm of the anther, which was cleared and stained with Calcofluor White to stain the cell wall components. The stained anthers were also visualized using Lightsheet microscopy. Cells of the epidermis were elongated in the inner anther locules in the WT, while the epidermis cells were shorter in the *mir2118* mutants (Fig. 2j–l). miR2118 was highly expressed in the

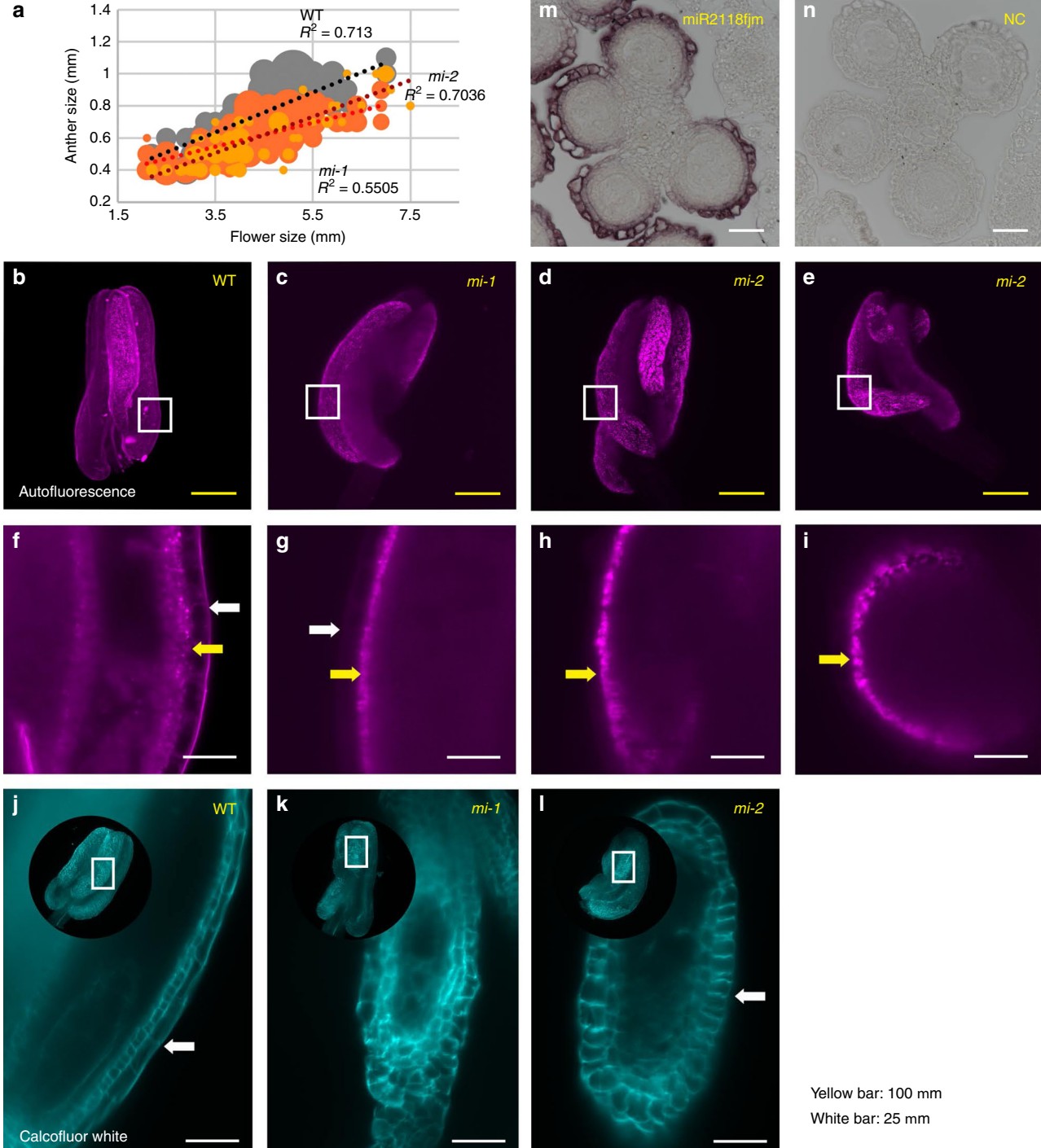

**Fig. 2 *mir2118* exhibits defects in epidermis development in anther wall layers. a** Correlation between anther length and flower size in WT (gray), *mi-1* (orange), and *mi-2* (light orange). **b–i** Autofluorescence images of 0.5 mm anthers of WT, *mi-1*, and *mi-2*, as captured by Lightsheet microscopy. An anther comprises four locules: two outer and two inner locules. Outer locules were curved at Stage 2 in *mi-1* (**c**) and *mi-2* (**d, e**). Enlarged images of the anther wall (**f–i**). Autofluorescence was strongly detected in the epidermis of the WT anther wall, while these epidermis signals were not evident in the *mir2118* mutants. White and yellow arrows represent the epidermis and tapetum layers, respectively. **j–l** Images of 0.4 mm anthers of WT, *mi-1*, and *mi-2*, as stained with Calcofluor White and captured by Lightsheet microscopy. Cells of the epidermis in inner locules were elongated in WT, and not elongated in the *mir2118* mutants. White arrows indicate epidermis cells in inner locules. Yellow and white bars indicate 100 and 25 μm, respectively. **m**, **n**. In situ hybridization of miR2118fjm and negative control (NC) probes (miRCURY LNA miRNA detection probe; see Supplementary Table 4) using anthers of 2.0–2.5 mm inflorescence, at Stages 1 and 2. miR2118fjm was strongly expressed at the epidermis in the anther walls. White bar represents 25 μm. Excitation wavelength/fluorescence are 561/LP585 nm for autofluorescence (**b–i**) and 405/420–470 nm for Calcofluor White (**j–l**). Maximum intensity projection was used for the anther overview.

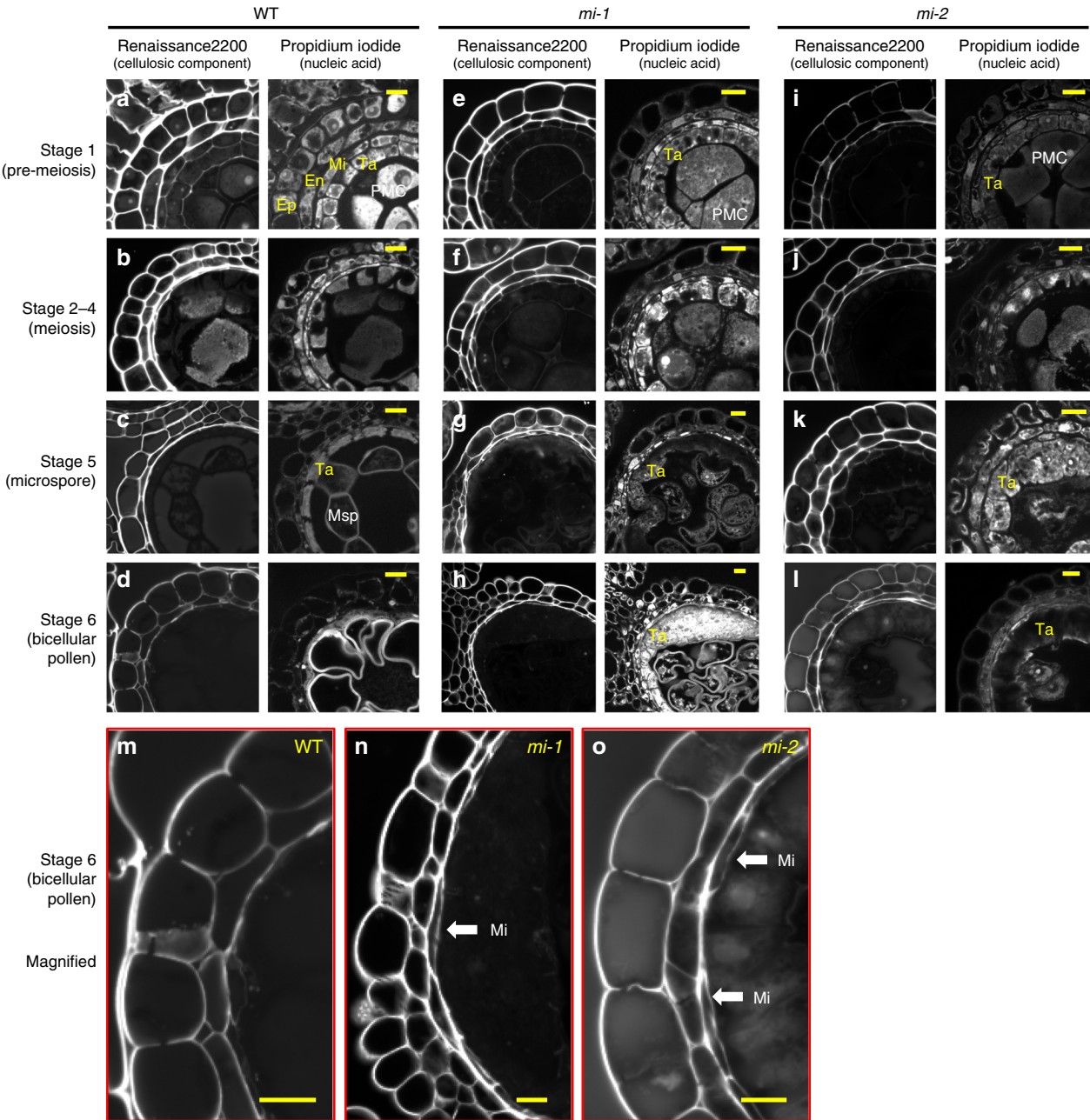

**Fig. 3 miR2118 is involved in the maturation of the anther wall during post meiosis. a–l** Cross-section images of anthers of WT (**a–d**), *mi-1* (**e–h**), and *mi-2* (**i–l**). Panels depict anthers that were stained with Renaissance 2200 (left) and propidium iodide (right). Stage 1 is pre-meiosis, or primordial germ-cell initiation (**a**, **e**, and **i**); Stages 2–4 are during meiosis (**b**, **f**, and **j**); Stage 5 is the microspore development stage (**c**, **g**, and **k**); and Stage 6 is the bicellular pollen stage (**d**, **h**, and **l**). Tapetum defects were observed in *mi-1* and *mi-2* at Stages 5 and 6 after meiosis (**g**, **h**, **k**, and **l**). **m–o** Enlarged images of the anther wall that was stained with Renaissance 2200 of **d**, **h**, and **l**. Middle layers were retained at later stages, even at Stage 6, in both *mi-1* and *mi-2* compared to the Mi-collapse in the WT anther wall at Stage 5. Ep: Epidermis, En: endothecium, Mi: middle layer, Ta: tapetum, PMC: pollen mother cell, Msp: microspore. The white arrows indicate Mi. Yellow bars represent 10 μm.

epidermis of the WT anther walls at Stage 2 (Fig. 2m, n)[25], which coincides with the phenotypes of weak autofluorescence and short cell sizes of the *mir2118* epidermis. These results indicate that *mir2118* is required for the longitudinal elongation of inner and outer anther locules via epidermis development.

**The involvement of miR2118 in the maturation of anther wall.** Next, we investigated anther wall layers from pre-meiotic stages to microspore development stages by histochemical staining, in which the cellulosic component was stained using Renaissance

2200, and RNA/DNA were stained using propidium iodide[26]. During Stages 1–4, the development of both pollen mother cells/meiocytes and somatic anther wall layers in the outer locule were comparable between WT and *mir2118* (Fig. 3). However, during Stages 5 and 6 after meiosis, *mir2118* exhibited abnormalities in tapetum maturation in curled anthers. Although tapetum layers became thinner in the WT anther walls, due to PCD initiation in propidium iodide staining, the tapetum layers remained thick in *mir2118*, even in the microspore stages (Fig. 3c, g, k). Moreover, the middle layer remained even at the bicellular pollen stages

**Table 1 Developmental stages of anthers.**

| Stages | Anther (mm) | Inflorescence (mm) | Germ cell development | Anther wall development |
|--------|-------------|--------------------|-----------------------|-------------------------|
| Stage 1 | 0.4 | 2 | Pre-meiosis (primordial germ-cell initiation) | Four layers (Ep, En, Mi, Ta) |
| Stage 2 | 0.5 | 2.5–4.0 | Leptotene–Zygotene | Ta differentiation |
| Stage 3 | 0.6 | 2.5–4.0 | Pachytene–Diplotene | Ta differentiation |
| Stage 4 | 0.6–0.7 | 2.5–4.0 | Meiotic division | Ta differentiation |
| Stage 5 | 0.8–0.9 | 5.0–6.0 | Microspore | Ta programmed cell death, Mi disappearance |
| Stage 6 | 0.9–1 | 7 | Bicellular pollen | Ta degeneration |

*Ep* epidermis, *En* endothecium, *Mi* middle layer, *Ta* tapetum.

(Stage 6), while it generally disappeared in the WT (Fig. 3m–o). These results suggest that miR2118 is essential for the maturation of anther wall layers

A previous study demonstrated that homologous chromosome synapsis during early meiosis in rice male germ cells requires the AGO protein MEL1, which binds to 21-nt phasiRNAs[10]. To examine how *mir2118* affects male meiosis, pollen mother cells from Stages 1 to 3 were compared between WT and *mir2118*. The leptotene–zygotene marker protein PAIR2 and the pachytene marker OsZEP1 were localized on meiotic chromosomes at pachytene Stage 3 in *mir2118* pollen mother cells, similar to the localization in WT (Supplementary Fig. 5). Subsequently, the synapsis process of the homologous chromosomes was completed in *mir2118* with elongation of OsZEP1 localization, which was not observed in *mel1* mutants (Supplementary Fig. 5)[10]. Overall, our data indicated that miR2118 was primarily involved in the development of anther walls, rather than germ-cell development in early meiosis, in contrast to the 21-nt phasiRNAs that were bound by MEL1 in germ cells.

**Identification of miR2118-dependent phasiRNAs in the anther.**
To understand the role of miR2118 in small RNA biogenesis during rice reproduction, we performed transcriptome analysis of small RNAs that were extracted from the 0.5 mm anthers at Stage 2, from WT, *mi-1*, and *mi-2* (two replicates for each sample; Supplementary Table 3). Our data confirmed that the expression levels of eight mature miR2118 families in the deleted region, including miR2118bn, d, fjm, g, h, k, i, l, and o, were reduced in both anthers in *mir2118* (Supplementary Fig. 6a, b). We found in WT that 39.1% of the anther small RNAs were 21-nt phasiRNAs, while 9% were 24-nt phasiRNAs (Fig. 4a, b). Importantly, the total counts of the anther 21-nt phasiRNAs were decreased in both *mi-1* and *mi-2* compared to in the WT (Supplementary Fig. 6c), which indicates that the 21-nt phasiRNAs were highly enriched in the anther and the loss of the miR2118 function causes defects in the production of 21-nt phasiRNAs

To identify phasiRNA clusters that are specific to anther walls and that are putatively regulated by miR2118, we conducted a comparative analysis on 21-nt phasiRNA clusters using the *PHASIS* program[27]. In total, 1345 21-nt phasiRNA clusters were detected in the anther (Fig. 4c; Supplementary Data 1). Moreover, we detected 1072 clusters of phasiRNAs that interact with MEL1, which localized in germ cells and not in somatic anther walls (Fig. 4c)[10]. By subtracting the overlapping MEL1-phasiRNA clusters (germ-cell phasiRNAs) from anther phasiRNA clusters, we defined the remaining 532 phasiRNA clusters as anther wall-specific phasiRNA clusters (Fig. 4c). In parallel, we identified 648 phasiRNA clusters that were significantly decreased in *mir2118*, suggesting that their production likely depend on miR2118 (Fig. 4d–f, Supplementary Data 1). In the miR2118-dependent phasiRNA clusters, 21-nt phasiRNAs were decreased in *mir2118* during male reproduction, from Stages 1 to 6, while the expression of the

precursors was elevated in *mir2118* (Fig. 4d, e; Supplementary Fig. 7). Eventually, 320 clusters that overlapped between 648 miR2118-dependent and 532 anther wall-specific clusters were extracted (Fig. 4g) and considered as miR2118-dependent anther wall-specific phasiRNA clusters.

In maize, 24-nt phasiRNAs are generated in the meiotic stage by the processing of miR2275 and DCL3b/DCL5 after the 21-nt phasiRNA production at the pre-meiotic stage[13]. In the *mir2118* mutant, 88.9% (60 out of 70) 24-nt phasiRNA clusters did not show significant changes in phasiRNA production (Supplementary Fig. 8), suggesting that miR2118 has a minor impact on 24-nt phasiRNA production. This is consistent with the normal meiosis progression in *mir2118* (Supplementary Fig. 5).

**U-rich phasiRNAs in miR2118-dependent anther wall clusters.**
AGO proteins are known to bind to the small RNAs with the specific nucleotide at the 5′ end[28,29]. Rice MEL1, categorized as an AGO5 subfamily group in the phylogenetic tree, preferentially binds to 21-nt phasiRNAs with C at the 5′-terminal position (Fig. 5a, b)[10]. To the contrary, in phasiRNAs from the 320 miR2118-dependent anther wall clusters (Fig. 4g), the frequency of the U nucleotide at the 5′ terminal was 40.7% higher than the frequencies of the other nucleotides (Fig. 5c, d). The U content was also enriched at the 3′ terminal in miR2118-dependent anther wall phasiRNAs. In the phasiRNAs of the 1345 anther clusters, the 5′-terminal sequence compositions of C (35%) and U (33.1%) were comparable (Fig. 5e, f), which may reflect the proportion of germ cells with first C-phasiRNAs and anther walls with first U-phasiRNAs. Anther phasiRNAs also exhibited enriched G at the 19th position (Fig. 5e, f). The 2-nt 3′ overhang that is formed by DCL processing was found to cause the 19′ position being G, which is complementary to the 5′-terminal C in the phasiRNA duplex[30]. These results indicate that the U-rich nucleotides are characteristic to miR2118-dependent phasiRNAs in the anther wall, which is evidently distinct from C-phasiRNAs in the germ cells.

**The site-specific expression of *OsAGO1b/d* in the anther wall.**
To explore the AGO proteins involved in miR2118 and anther wall phasiRNA biogenesis, we performed proteome analysis using total proteins extracted from the 0.5 mm anthers (Stage 2) of WT, *mi-1*, and *mi-2* (three replicates for WT, two replicates for *mi-1* and *mi-2*). We found that MEL1 and AGO1d were significantly decreased in both *mi-1* and *mi-2* ($p$ value < 0.05, Fig. 6a, b). Reductions in OsAGO1b (in *mi-1*) and OsAGO1c (in *mi-2*) were also detected among the OsAGO family members ($p$ value < 0.05; Fig. 6a, b). Western blot analysis using MEL1 antibodies demonstrated a reduction of the protein in *mir2118* at Stages 2 and 3, which was validated by the proteome results (Fig. 6c). Besides the AGO proteins, TDR1 (TAPETUM DEGENERATION RETARDATION1) and TIP2 (TDR INTERACTING PROTEIN2), were also decreased in *mir2118* (Supplementary Data 2), which are basic helix–loop–helix transcription factors

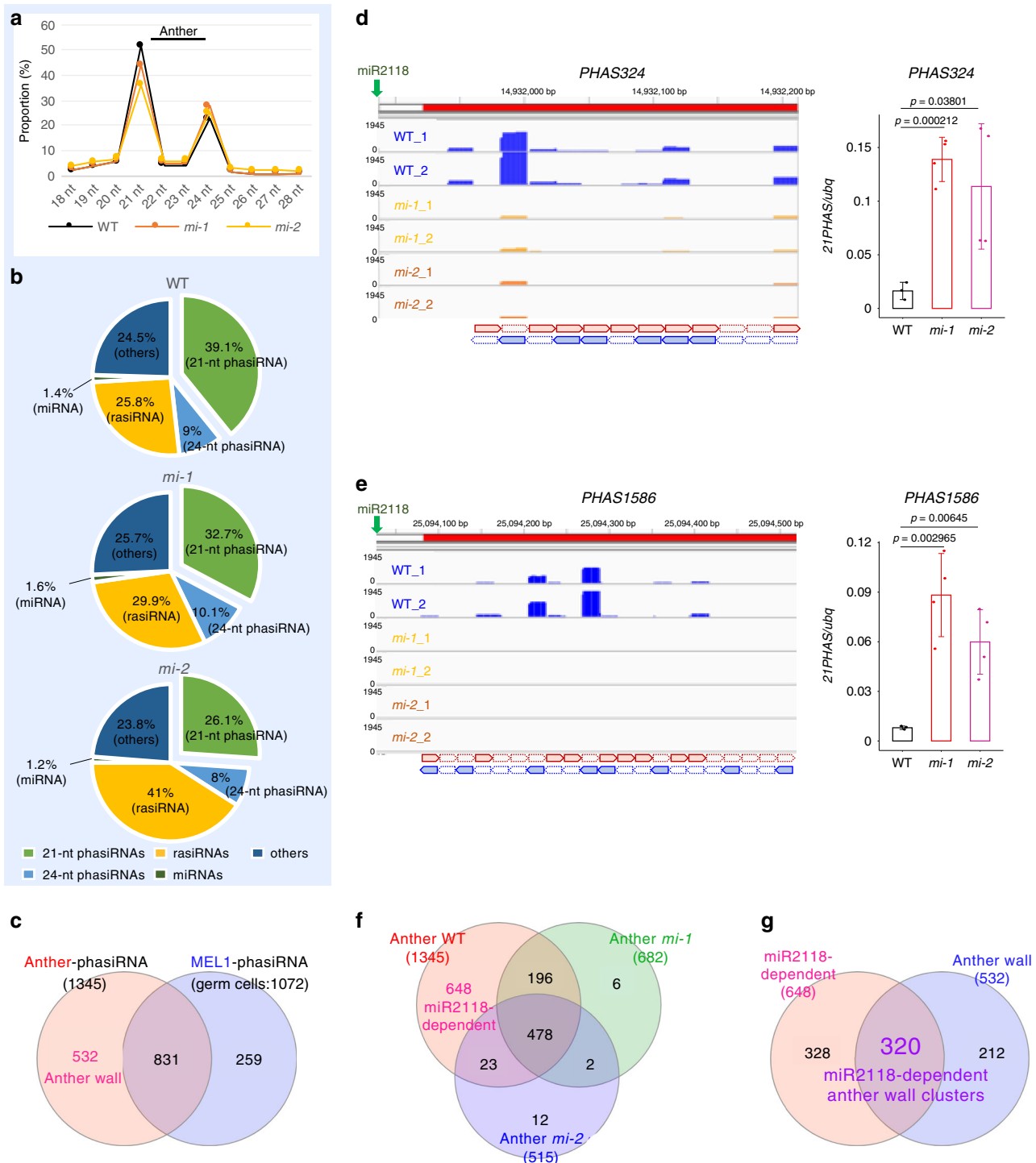

**Fig. 4 Identification of miR2118-dependent phasiRNAs in the anther wall. a** Size distribution of small RNAs from the WT (black line) and *mi-1* (orange line) and *mi-2* (yellow line) lines that were isolated from 0.5 mm anthers. **b** Pie charts summarizing the source of anther small RNAs from WT, *mi-1*, and *mi-2*. Anther small RNAs were mainly 21 nucleotides (nt) and derived from the lincRNAs/*PHAS* loci. **c** Venn diagram of overlapping anther 21-nt phasiRNA clusters and MEL1 21-nt phasiRNA clusters (germ-cell clusters). **d**, **e** Small RNA-seq reads of WT, *mi-1*, and *mi-2* with two replicates that were mapped to the *21PHAS* loci. Schematic alignment of 21-nt phasiRNAs is illustrated at the bottom by thick horizontal arrows with red (sense) and blue (antisense). Right panels depict the qPCR analysis of *PHAS324* and *PHAS1586*, lincRNAs, in 0.5 mm anthers at Stage 2. Bar represents mean ± SD (*N* = 3 or 4). Student's *t* test. **f** Venn diagram of overlapping phasiRNA clusters that were identified in the anthers of WT and *mi-1* and *mi-2* lines. **g** Venn diagram shows overlapping miR2118-dependent clusters (648; **f**) and anther wall-specific phasiRNA clusters (532; **c**). The 320 overlapping clusters were considered miR2118-dependent phasiRNA clusters in the anther wall.

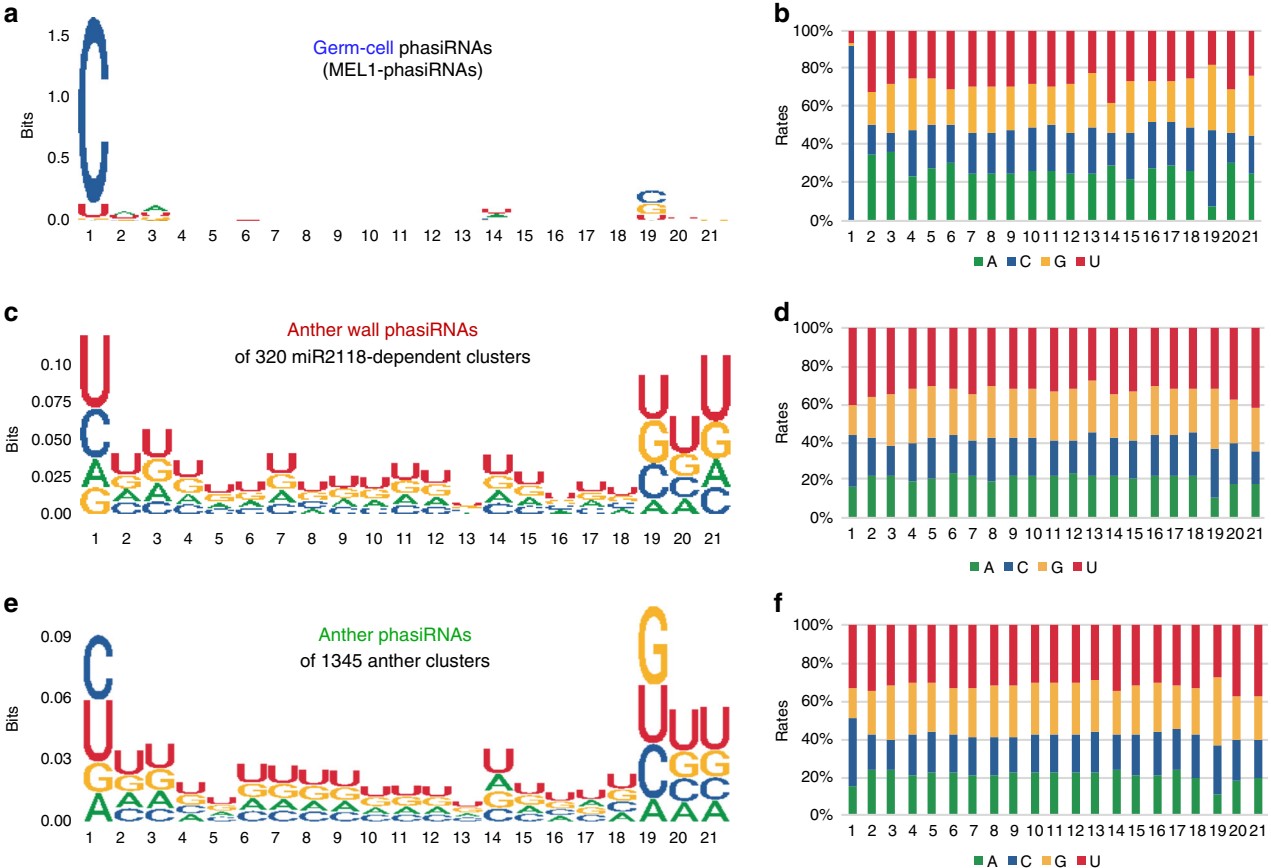

**Fig. 5 U-rich phasiRNAs are enriched in the miR2118-dependent anther wall clusters. a, c, e** Relative nucleotide bias at each position of the 21-nt phasiRNAs. **b, d, f** Relative frequency of each nucleotide in the 21-nt phasiRNAs. The 5′ terminals of phasiRNAs that interact with MEL1 are mainly "C" (**a**, **b**), while miR2118-dependent phasiRNAs from the 320 anther wall phasiRNA clusters are "U" rich, especially at the 5′ and 3′ ends (**c**, **d**). In the whole anther phasiRNAs, the 5′-terminal sequence composition of "C" and "U" are comparable (**e**, **f**).

required for the specification of the inner anther wall layers and tapetum differentiation[20]. These results suggest that the miR2118 affects the anther wall-specific transcription factors via anther differentiation directly or indirectly.

There are four AGO1 class proteins, OsAGO1a, OsAGO1b, OsAGO1c, and OsAGO1d, out of 19 AGO proteins in rice[31,32]. In the OsAGO1 subgroups, *OsAGO1a* is expressed during vegetative stages, while *OsAGO1b*, *OsAGO1c*, and *OsAGO1d* are highly expressed during reproductive stages (Supplementary Fig. 9). In addition, their expressions were not significantly affected in *mir2118* anthers compared to the expression of WT (Supplementary Fig. 9). To elucidate the mRNA localization of *OsAGO1b*, *OsAGO1c*, and *OsAGO1d*, in situ hybridization was performed using anthers at Stages 1 and 2. As reported, *MEL1* was expressed in pollen mother cells at Stage 1 (Fig. 6j)[22]. We detected the expression of *OsAGO1b* and *OsAGO1d* in the anther wall cells (Fig. 6d–i), although *OsAGO1c* expression was not detected (Supplementary Fig. 10). *OsAGO1b* was expressed in the middle layer and endothecium at Stage 2 (Fig. 6h). *OsAGO1d* transcripts were enriched in the tapetum layer, middle layer, and endothecium, and slightly in the epidermis, which is consistent with the expression pattern of *OsAGO1d* in the previous study (Fig. 6d, e)[25]. It has also been reported that *Arabidopsis* AtAGO1 and rice OsAGO1a, b, and c bind to the U-5′ terminal of microRNAs[28,33]. Thus, these results suggest that OsAGO1b and OsAGO1d, which are highly expressed in the anther wall, may be involved in U-miR2118 and/or U-phasiRNAs biogenesis during anther wall development.

## Discussion

We demonstrated in this study that the deletion of the miR2118 cluster in the rice genome induces severe defects in anther wall development and sterility via the reduction of 21-nt phasiRNA production. Importantly, we identified novel U-rich 21-nt phasiRNAs as miR2118-dependent small RNAs in the anther wall (Figs. 1–5). It remains unclear whether the 21-nt anther wall phasiRNAs with specific nucleotides at the 5′ terminals are sorted to AGOs, which results in the enrichment of the nucleotides (Fig. 5). In this regard, we found that *OsAGO1b* and *OsAGO1d* are highly expressed in the anther wall, and that the loss of miR2118 affects protein levels of OsAGO1b, OsAGO1d, and MEL1 (Fig. 6). There are two possibilities to explain the reduction of AGO1b, AGO1d, and MEL1 proteins in *mir2118* mutants: (1) the reduction of miR2118 or phasiRNAs may cause the destabilization of AGO1b/AGO1d/MEL1 proteins in the absence of miR2118/phasiRNAs; or (2) defects of anther wall development in the *mir2118* may reduce the anther wall cell mass, which then reduces the relative amounts of AGO1 proteins. Reduced AGO1b/d proteins with no difference of the mRNA levels in *mir2118* mutants (Supplementary Fig. 9) may support the first hypothesis, in which the proteins are associated with the anther wall U-rich phasiRNAs or U-miR2118 family members. Overall, we propose a model of U-rich phasiRNAs biogenesis that is induced by miR2118, along with anther wall-specific AGO1b/d during anther wall development (Fig. 6k). In contrast, MEL1 interacting with C-phasiRNAs regulates synapsis formation in pollen mother cells[10].

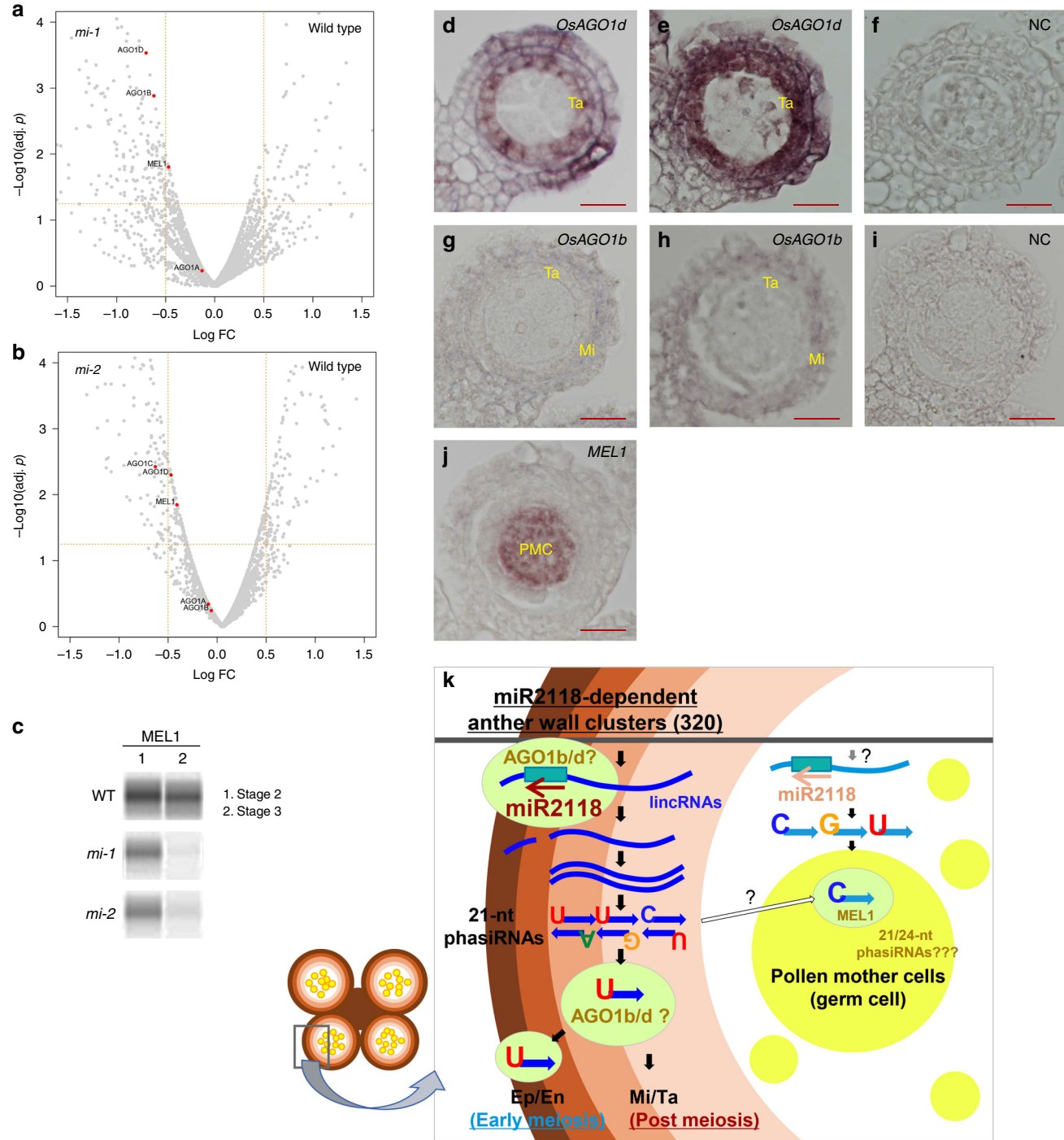

**Fig. 6 *OsAGO1b/OsAGO1d* is highly expressed in the anther wall. a**, **b** Volcano plots of proteome analysis in 0.5 mm anthers (Stage 2) of WT, *mi-1*, and *mi-2* (WT, three biological replicates; *mi-1* and *mi-2*, two biological replicates). **c** Western blot analysis of MEL1 in the anther. Total proteins that were extracted from 0.5 mm anthers (lane 1, Stage 2) and 0.6 mm anthers (lane 2, Stage 3). MEL1 protein is reduced in *mi-1* and *mi-2* at Stages 2 and 3. **d–j** *In situ* hybridization of *MEL1*, *OsAGO1b*, and *OsAGO1d* in the anther. *OsAGO1d* expression is abundant in the anther wall, especially at the tapetum (Ta), middle layer (Mi), and endothecium (En) regions. *OsAGO1b* localization was more enriched in Mi and epidermis (Ep) than Ta. *MEL1* is specifically expressed in pollen mother cells (PMC). **d**, **g**, **j** are at Stage 1, and **e**, **h** are at Stage 2. NC: Negative controls with the sense strand of *OsAGO1d* (**f**) and *OsAGO1b* (**i**) at Stage 2 . Bars = 20 μm. **k** Model of cell-linage-specific phasiRNA production during anther development. During early meiosis, miR2118 expression is increased in the epidermis, and promotes the production of 21-nt phasiRNAs in the anther wall. U-phasiRNAs, whose production is dependent on miR2118, may be sorted into OsAGO1. OsAGO1b/d-miR2118 and/or OsAGO1b/d-U-phasiRNAs may function in anther wall development during early meiotic stages. In contrast, in PMC, C-phasiRNAs that interact with MEL1 regulate meiosis progression. After meiosis, miR2118 may further act to regulate the maturation of Ta and Mi.

In animals, the gonad-specific AGO proteins of the PIWI family bind with 23- to 30-nt PIWI-interacting RNAs (piRNAs)[34]. Three AGO proteins, PIWI, Aubergene (Aub), and AGO3, that bond to piRNAs are involved in the suppression of transposable elements in the *Drosophila* germline. PIWI is crucial for the phased piRNA production in ovarian somatic cells, while secondary piRNAs are amplified via PIWI/Aub/AGO3 in ovarian germ cells[35]. Thus, site-specific AGO-small RNA complexes are essential for reproductive development in both animals and plants. Further studies would elucidate both the site-specific molecular mechanism and the significance of AGO/phasiRNAs/miR2118 in plants.

*mir2118* mutants exhibited abnormal epidermis development during early reproduction, and then abnormalities of tapetum and middle layer maturation during post meiosis (Figs. 2 and 3), suggesting that miR2118 may have dual functions in the specific anther wall cell layers during anther development. Alternatively, the defects of the tapetum/middle layers in the *mir2118* mutants are secondary effects of the abnormal differentiation of outer layers during pre-meiosis. In maize, OUTER CELL LAYER 4 (OCL4), an HD-ZIP IV-type transcription factor, represses extra periclinal divisions in the epidermis and endothecium and regulates miR2118 expression and 21-nt phasiRNAs production[13,36,37]. In rice, *MULTIPLE SPOROCYTE* (*MSP1*), which is abundant in tapetum, regulates the sporogenous cell number. *msp1* exhibits defects in epidermis elongation, as demonstrated with the *mir2118* phenotype (Fig. 2)[38]. Cell-to-cell transportation of siRNAs is important for successful reproduction after fertilization[39]. An interaction between the soma and germ or a non-cell autonomous regulation via phasiRNAs and transcription factors may be key to fully explaining the anther development mechanism.

The *21PHAS*/lincRNAs sequences are diverse among even Poaceae[10], which indicates a functionally divergent species-specific evolution of *21PHAS*/lincRNA. In contrast, 22-nt miR2118 cleavage and DCL processing in phasiRNA biogenesis are widely conserved in land plants, angiosperm, and gymnosperm[16]. These results indicate that high functional conservation of miR2118 parallels the emergence of fast-evolving *PHAS* loci in various plant genomes, which may be adaptive under various environments. It has been reported that defects in the phasiRNA pathway result in temperature-sensitive sterile or photoperiod-sensitive sterile phenotypes[24,40]. Intriguingly, *mir2118* showed both male and female photoperiodic sterility (Fig. 1). However, no significant changes were observed in the expression levels of well-known photoperiodic factors (*PMS1T* and *Hd1*, *Ehd1*, *Hd3a*, *RFT1*, *OsMFT1*, and *OsMADS50*) (Supplementary Fig. 11), which indicates that the sterility in *mir2118* is independent of these photoperiodic flowering pathways. Accordingly, the combination of miR2118 and *21PHAS*/phasiRNAs may be a reproductive strategy to prosper offspring, including a novel adaptive mechanism for an environmental response.

## Methods

### Plant materials and growth conditions
*Japonica* rice "Nipponbare" was used as a WT plant in the analysis, except for the segregation and fertility tests that are represented in Supplementary Fig. 2, in which segregating WT and heterozygous siblings of *mir2118* were used as controls. Plants were grown in growth chambers at 70% humidity either under SD conditions with daily cycles of 10 h of light at 30 °C and 14 h of dark at 25 °C or LD conditions with 14 h of light and 10 h of dark. In the case of plants growing under SD conditions, plants were grown for 6 weeks under LD conditions and then transferred to the SD conditions until harvest. *mi-1* mutant was backcrossed once with "Nipponbare" (BC1) and used for further studies.

### Gene targeting construction and transformation
The pZH_gYSA_MMCas9 and pU6gRNA-oligo vectors were used as materials[41]. For editing of the miR2118 loci, 5′-CAATAGGAATGGGAGGCATC-<u>AGG</u> (PAM)-3′ is used for the target

sequence of the CRISPR/Cas9 vector as single guide RNAs. First, the oligos for guide RNAs are cloned to pU6gRNA-oligo vectors. Second, OsU6 promoter::gRNA is cut with *Sce*I and is cloned to the pZH_gYSA_MMCas9 binary vector with hygromycin phosphotransferase selection[41]. Transgenic rice plants were generated by *Agrobacterium*-mediated transformation of rice calli ("Nipponbare")[42].

### RNA extraction, qPCR, and miRNA/phasiRNA-qPCR
Total RNAs from each stage of the 0.4, 0.5, 0.6, 0.7, 0.8, and 0.9 mm anthers (quantitative PCR (qPCR)) and from rice leaf blades, leaf sheaths, roots, and 1–5 cm panicles (for developmental expression analysis) were extracted using TRIzol® Reagent (Invitrogen). qPCR was performed using the KAPA SYBR FAST qPCR Kit (Nippon Genetics)[43]. miRNAs/phasiRNA-qPCR was conducted with the small RNA Quantitation System QuantiMir RT Kit (System Biosciences) according to the manufacturer's instructions. Primers used for the qPCR are listed in Supplementary Table 4.

### Small RNA-sequencing
For small RNA-seq analysis, total RNAs were extracted from 0.5 mm anthers (Stage 2) using TRIzol® Reagent (Invitrogen). Two biological replicates were prepared for "Nipponbare" (NB), *mi-1* (T2BC1), and *mi-2* (T3) for small RNA-seq. Illumina RNA-Seq libraries (150 bp paired end) were prepared and sequenced using Illumina Hiseq 2500 (GENEWIZ, Japan).

### Small RNA-sequencing data analysis
Sequencing reads were trimmed using TrimGalore (http://www.bioinformatics.babraham.ac.uk/projects/ trim galore) to remove adapter sequences and sequencing bias with the following parameters: stringency = 5, quality = 20, and three prime clip R1 = 60. After trimming, reads that ranged from 18 to 30 bp were used. To annotate the small RNA, the perfectly matched reads were mapped onto the rice genome IRGSP1.0 (https://rapdb.dna.affrc.go.jp) with TopHat (2.1.1) (https://sourceforge.net/projects/bowtie-bio/files/bowtie2/), and the datasets of miRNA from miRbase (http://www.mirbase.org/ftp.shtml) and repeat data from RAP-DB (https://rapdb.dna.affrc.go.jp/) were used.

To identify 21-nt and 24-nt phasiRNA clusters, 18- to 30-nt trimmed small RNAs were analyzed using *PHASIS*[27] with default parameters. Significant clusters ($p$ value $\leq 1e^{-5}$) from the replicates were merged. Overlapping clusters among the samples were extracted using the phasmerge function in *PHASIS*. Annotations of MEL1-associated 21-nt-phasiRNA clusters that were provided in the study by Komiya et al.[10] were also used to perform comparisons with *PHASIS*-detected clusters[10]. Position weight matrices that correspond to consensus sequences of 21-nt phasiRNAs were calculated using custom R scripts and visualized using ggseqlogo[44].

### Whole-genome sequencing and detection of genomic variation
Illumina whole-genome sequencing libraries (150 bp paired end) were prepared and sequenced using Illumina Hiseq 2500 (GENEWIZ, Japan for WT, *mi-1* and *mi-2*, or AnnoRoad, China for *mi-3*). To detect the variation between the *mir2118* mutant lines, we performed the Basic Variant Detection of CLC Genomics Workbench 11.0.1 procedure.

### In situ hybridization
Inflorescences were fixed in FAA solution (30% ethanol, 1.85% formaldehyde, and 5% acetic acid) overnight at 4 °C and then dehydrated in a graded ethanol series. Ethanol was replaced with *t*-butyl alcohol and the samples were embedded in Paraplast Plus (Merck)[45]. The probes of the gene regions were synthesized using the Digoxygenin Labeling Kit (Roche, USA) following the manufacturer's instructions. The primer sets for the probes and microRNAs that were modified with LNA and DIG at the 3′ and 5′ terminals are listed in Supplementary Table 4. Cross-sections of the tissues with 8 μm thickness were prepared using a rotary microtome. A part of the sections was stained with 0.1% (w/w) toluidine blue to identify the anthers. Sections mounted on slides were dewaxed with Lemosol (Fujifilm-Wako). Slides were incubated with 0.5 μg/ml proteinase K (Sigma) for 15 min at 37 °C, and then acetylated with 1.5% (v/v) triethanolamine in a solution of 0.25% (v/v) concentrated hydrochloric acid and 0.25% (v/v) acetic hydride. Gene probes (0.8 μg/ml) or 25 nM microRNA probes were used for hybridization overnight at 55 °C. The slides were washed twice with 50% (v/v) formamide in 2× saline-sodium citrate (SSC) for 30 min at 55 °C and incubated with the buffer containing 10 μg/ml RNaseA for 30 min at 37 °C to remove the unhybridized RNA probes. Next, the slides were washed twice with 2 × SSC and 0.2 × SSC for 30 min at 55 °C. The blocking steps and detection of hybridized transcripts, which required anti-digoxigenin antisera conjugated to alkaline phosphatase (Roche Anti-Digoxigenin-AP, NBT/BCIP), were performed following the manufacturer's protocol. The signal was detected after 3 to 16 h of incubation at 30 °C[46].

### Histochemical staining using anthers and immunofluorescent cytological analysis
Anthers fixed in 4% paraformaldehyde were used for immunofluorescent cytological analysis and histochemical staining[47]. For morphological analysis of the anthers (Fig. 2), 4% paraformaldehyde-fixed anthers were transferred to the ClearSee solution[48] and cleared at room temperature for one day. The anthers were stained in 0.1% (w/v) Calcofluor White in ClearSee solution for 30 min[49] and washed thrice in ClearSee. Specimens were embedded with 0.1% agarose gel in a

glass capillary. Image was captured using a Lightsheet illumination microscope with 20× Plan Apochomat lens (Carl Zeiss Z.1). For imaging of autofluorescence, anthers fixed in 4% paraformaldehyde and washed in phosphate-buffered saline were embedded with 0.1% agarose gel in glass capillary and images were captured under Lightsheet system with the 20× lens.

For the cross-section of the anthers (Fig. 3), the samples fixed in 4% paraformaldehyde were embedded in Technovit 7100 (Kulzer) following the manufacturer's instructions. Technovit samples were sliced into 8–10 μm sections using rotary microtome. The sliced samples were stained with Renaissance 2200 (Renaissance Chemicals) and were washed thrice in distilled water. The samples were mounted with Vectashield mounting medium with propidium iodide (Vector Laboratories)[26].

For immunostaining (Supplementary Fig. 5), subcellular localization of PAIR2 and ZEP1 was observed by indirect immunofluorescent staining of pollen mother cells (PMCs). Anthers from fixed inflorescence were incubated in an enzyme cocktail that contained 2% cellulase Onozuka-RS (Yakult Honsha, Japan), 0.3% Pectinase (Tokyo Chemical Industry), and 0.5% Macerozyme-R10 (Yakult Honsha) in PME buffer (50 mM PIPES, 10 mM EGTA, 5 mM $MgSO_4$, pH 6.9), for 10 sec at room temperature. The anthers were washed thrice with PME on a MAS-coated microscope slide and squashed in distilled water by a needle to release PMCs. Then, PMCs were blocked with 3% bovine serum albumin (BSA) in PME for 60 min, followed by a PME wash for 5 min. The PMCs were incubated overnight at 4 °C with rabbit anti-PAIR2 antibody and rat anti-ZEP1 diluted 1/2000 and 1/1000 with 3% BSA in PME. After washing thrice with PME for 5 min, the slide was incubated in a dark chamber for 3 h at room temperature with Alexa Fluor 488-conjugated anti-rabbit IgG (Invitrogen, A-11034) and Alexa Fluor 647-conjugated anti-rat IgG (Invitrogen, A-21247) diluted 1/200 with 3% BSA in PME, followed by washing thrice with PME for 5 min each. Then, samples were mounted in Vectashield mounting medium with DAPI (4′,6-diamidino-2-phenylindole; Vector Laboratories)[10,47]. Images for both immuno- and histochemical staining of resin cross-sections were captured with a confocal microscope (LSM780; Carl Zeiss).

The PAIR2 complementary DNA (cDNA) was amplified by PCR with a primer pair of 5′-CACCATGGTGATGGCTCAGAAGACGAAG-3′ and 5′-TCACTGA ACTTGAACTTGAACTTGGGAC-3′. An N-terminal region of ZEP1 cDNA encoding the 17–436 amino acid region was amplified by PCR with a primer pair of 5′-CACCTTAGAAGTACTGTTTCAGGGCCCGTCTCTCGCTGGATCCACC-3′ and 5′-TCATTCAGCAGATCTAGAATCCTCC-3′. The amplicon of PAIR2/ZEP1 was cloned into each pENTR/D-TOPO, reinserted into pDEST17 (Gateway system, Invitrogen), and independently transferred into Escherichia coli strain BL21-AI (Invitrogen). 6xHis-tagged PAIR2/ZEP1 peptide was purified using HisTrapFF Crude Kit (GE Healthcare) according to the manufacturer's instruction. The purified peptide was immunized to rabbit for PAIR2 or to rat for ZEP1, and the antiserum was used for immunostaining[23,47].

**Mass spectrometry sample preparation**. The 0.5 mm anthers of WT, mi-1, and mi-2 mutants were grinded and mixed with extraction buffer (150 mM NaCl, 50 mM Tris-HCl (pH 7.0), 0.1% Tween-20, 10 % glycerol, 1 mM dithiothreitol (DTT), 1 mM Pefabloc SC (Roche), 1× Complete Protease Inhibitor Cocktail). After centrifugation twice and removing the debris, total proteins were extracted[50]. Two to three replicates were prepared for each sample.

The protein sample was reduced with DTT and then alkylated with iodoacetamide and digested overnight using Lys-C/Trypsin combo (1:50, enzyme to protein; Promega). After terminating the digestion with 1% trifluoroacetic acid, the peptide mixture was cleaned with desalting C18 tips (StageTip, Thermo Fisher Scientific), and subsequently vacuum-dried and re-suspended for liquid chromatography-mass spectrometry (MS) analysis.

**MS data acquisition**. Data were collected using the Orbitrap-Fusion Lumos mass spectrometer (Thermo Fisher Scientific) coupled with the Waters nanoACQUITY Liquid Chromatography System (Waters Company). A trap-column (nanoAC-QUITY UPLC 2G-V/M Trap 5 μm Symmetry C18, 180 μm × 20 mm, Waters) and an analytical column (nanoACQUITY UPLC HSS T3 1.8 μm, 75 μm × 150 mm, Waters) were used for chromatographic separation of samples. Peptides were separated at a flow rate of 500 nl/min using a gradient of 1–32% acetonitrile (0.1% formic acid) over 60 min. The CHOPIN method was used along with the Orbitrap-Fusion mass spectrometer using Xcalibur (v.3.0; Thermo Fisher Scientific) and without modifications[51].

**MS data analysis**. Raw data files were searched against a composite target/decoy database using SEQUEST from Proteome Discoverer (PD, v.2.2, Thermo Fisher Scientific). The Oryza sativa L. subspecies japonica (Rice) protein database was downloaded from UniProt and was combined with the contaminants database (cRAP, https://www.thegpm.org/crap). MS2 spectra were searched with ±20 p.p.m. for the precursor ion mass tolerance and ±0.1 Da for the fragmentation mass tolerance using trypsin as an enzyme, a maximum of two missed cleavage sites, dynamic modifications for the oxidation of methionine, deamidation of glutamine and asparagine, and fixed modification for the carbamidomethylation of cysteine residues. Label-free quantification was performed based on peak intensity using

unique and razor peptides and was normalized using a specific protein amount (Trypsin).

**Western blot**. Total proteins were extracted from 0.5 mm (Stage 2) and 0.6 mm (Stage 3) anthers of the WT and mir2118 as noted above[50]. A western assay was performed using 2 μg of total protein per sample on Wes (protein simple). MEL1 antibodies (1/200 dilution) were used for the initial immune reactions. Two oligo peptides, MEL1-N (CVYGAPMPAAHHQGAYQ) and MEL1-C (GQAVAR-EGPVEVRQLPKC), were used to produce the MEL1 antibody, which has been described previously[10].

**Reporting summary**. Further information on research design is available in the Nature Research Reporting Summary linked to this article.

## Data availability

Whole-genome sequencing data and small RNA transcriptome data have been deposited in the DNA Bank of Japan (DDBJ), under the accession codes DRA010227 (whole-genome sequences) and DRA010228 (small RNA transcriptome). Proteome data were deposited to JPOST, under the accession code PXD015440. Source Data is provided as a separate file.

## Code availability

In-house R codes and bash scripts customized for analyzing data are available from the authors upon request.

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

## Acknowledgements
This work was supported by JST PRESTO (grant number JPMJPR17Q3), KAKENHI (grant numbers JP17H05608 and JP15H01476), and the Okinawa Institute of Science and Technology Graduate University, Japan to R.K. We thank Dr. Shinichi Nakagawa for their advice regarding in situ hybridization, Ms. Hinako Tamotsu for rice growth and genotyping, and Ms. Yasuka Shimajiri for the rice transformation. We thank Dr. Tim Hunt and all members of the Science and Technology Group and Plant Epigenetics units for their helpful discussions.

## Author contributions
R.K. conceived the study, conducted most of the data analysis, and wrote the manuscript. R.K. and S.A. performed mutant generation, small RNA-seq experiments, protein experiments, and immunostaining. K.K., S.A., and R.K. performed imaging experiments. N.T.L. performed bioinformatics analysis of phasiRNA clusters. A.V.B. performed proteome analysis. K.-I.N. provided the three antibodies. M.E. produced the CRISPR/CAS9 vector. H.I. supported the *mir2118* mutant phenotyping. H.S. and K.K. assisted in writing the manuscript and discussion.

## Competing interests
The authors declare no competing interests.
