## [Peer Review File · Nature Communications]

Reviewers' comments:

Reviewer #1 (Remarks to the Author):

In this manuscript, Araki et al. report an analysis of a rice mir2118 mutant generated using CRISPR-Cas9. The mutant appears to be a partial mir2118 mutant, as at least 4 (known) copies of miR2118 are not knocked out, and mature miRNAs derived from these loci are detected in anther and pistil. The mutant exhibits defects in both male and female reproductive development, causing male & female sterility. The developmental defects in anthers appear to be thicker tapetum and delayed death of middle layer. The authors also identified miR2118-triggered phasiRNAs in the somatic cells of anther and observed an enrichment of 5'-U in these phasiRNAs, distinct from the meiocyte-enriched phasiRNAs that are enriched for 5'-C. The analyses provide insights into the role of a subset of the miR2118 family in male (& female) reproductive development. However, the study is lacking some of the key data in support of the conclusions.

Below are a number of specific comments/suggestions that may help improve the manuscript.

Major issues:

1. There is no quantitative data (e.g., segregation ratios) provided for the genetic analysis of the mutant alleles.
2. The nature of the three mutant alleles is not clearly defined. For example, are the genomic sequences identical or different at the edited sites?
3. The deletion of a 16-kb genomic region using two guide RNAs is remarkable. However, it seems that the deleted region includes at least an intact protein-coding gene and a portion of another protein-coding gene, but the authors did not rule out (or discuss) these genes as potential causal genes of the mutant phenotypes.
4. On a related note, the authors state that "the phenotypic variations of the two alleles might originate from the gRNA sequences used for the deletion of the loci that have high homology with the miR2118 recognition sites in over 700 reproductive lincRNAs", implying editing at 21-PHAS loci; if this is the case, whole-genome sequencing of the two mutants, which is done in this study, may identify mutations in the 21-PHAS loci. However, this possibility is not discussed.
5. Some important information is missing in the figures/figure legends. In all figures except Supplementary Figure 1, it's not clear whether the mir2118 mutants are compared to their wild-type siblings or wild-type plants that are not their siblings.
6. There does not seem to be a clear set of hypotheses about the roles of AGO1b/AGO1d. The dysregulation of the AGO genes in the mir2118 mutant may simply suggest that they are regulated, either direct or indirectly, by miR2118 or the miR2118-triggered phasiRNAs, instead of being directly involved in miR2118/phasiRNA biogenesis as stated in the manuscript. The latter is in fact not well supported by the available data.
7. The writing of the manuscript needs polishing, perhaps by a native speaker. There are many issues in the writing that make it taxing to follow. For example, on page 7, I don't agree that the results indicate that mir2118 causes defects in the elongation of OUTER anther locules, as the change in the shape of the outer locules is likely a consequence of curled inner locules – as described in the manuscript.

Minor issues:

8. Supplementary Figure 1: labels of the y-axes are missing, so it's not clear what type of data is shown.
9. Supplementary Figures 2: It's not clear what error bars in panels b & c represent and how many plants are compared. The difference in plant height of mi-2 vs. wild type is not supported by any form of statistical test. In addition, images of whole plants are only shown for the mi-1 allele, but not for the mi-2 allele in which the dwarf phenotype is observed.
10. Supplemental Figure 4d looks like an interesting set of data showing overall reduction in phasiRNA abundance, but it's not clear whether it's the 21-nt or the 24-nt phasiRNAs. In addition, the data are derived from two replicates as indicated in the legend, but it's not clear whether they are mean or sum of the replicates.
11. Page 5: The authors state that "MEL1 is the only AGO protein known to bind phasiRNAs during reproduction" – it would be better to clarify that this is only in rice, as there are known AGO

proteins loading reproductive phasiRNAs in other plant species (e.g., maize).

Reviewer #2 (Remarks to the Author):

The manuscript describes the role of miR2118-dependent regulatory module in rice anther wall development by gene editing. The author integrated small RNA profiling, proteome, and 3D-histochemical analyses of miR2118 mutants to reveal the novel miR2118 functions and the biogenesis of miR2118-dependent U-phasiRNA/AGO1 subfamilies in the anther wall development. Overall, the experiments were well-designed and results were interesting. The data looked robust and well presented.

Comments/queries that need to be addressed:

1. Authors produced miR2118 mutants by genome editing and obtained three independent mutant alleles, mir2118-1, mir2118-2, and mir2118-3. Whether the authors conducted experiments such as southern blot to confirm these three mutants were actually independent mutant alleles? In Fig 1b, only mi-1 and mi-2 were presented, how about mi-3? Authors should add it to Fig 1b.
2. In manuscript, authors described that there was 16,230bp deleted in ir2118-1, mir2118-2, and mir2118-3(mi-1, mi-2, and mi-3) by whole-genome sequencing. So what is the PAM sequences to achieve such large fragment deletion? And what about the vector information?
3. Authors said microRNA2118 (miR2118) conserved among plants, is expressed at reproductive stages, causing the production of secondary phased small interfering RNAs (phasiRNAs) and it plays an important role in rice anther wall development. Does miR2118 has the similar function in other plants?
4. Authors found that there was no significant difference in plant height between WT and mi-1, while mi-2 showed a dwarf phenotype under SD condition. And is there any difference between WT and mutants under LD condition? Why did authors choose LD and SD conditions to investigate the role of miR2118 in rice anther wall development?
5. Authors classified rice anther development into six stages and selected Stage 2 to analyze the overall structure of anthers. And moreover, 0.5 mm anthers at Stage 2 were also used to perform the transcriptome analysis of small RNAs. So how about the different/differential expressions of small RNAs of the other five stages anthers between WT and miR2118 mutants? I suggested authors supply these data on the manuscript.
6. Expressions of encoding protein genes play important roles in rice anther development. Therefore, I wonder the differentially expressed genes between WT and miR2118 mutants. These data can be added to the manuscript if possible.
7. What's the role of AGO proteins in rice anther wall development? It should be clarified in the manuscript.
8. In present study, the inner mechanism of miR2118 in rice anther wall development is not so adequate and further research is suggested to be done to clarify the specific function of miR2118 in rice wall development.

Responses to reviewer's comments

First of all, we would like to thank the reviewers for their constructive and valuable comments with regard to our manuscript. We have indicated the major changes made in the revised manuscript, and provided further point-by-point responses to the reviewers' comments below.

Additional results and major changes

1. Long-deleted *mir2118* mutants are renamed as *mir2118-1-1*, *mir2118-1-2*, and *mir2118-1-3* in the main text (the reasons are as follows).
2. Identification of long-deleted sites and genomic variation of three *mir2118* mutants by whole genome sequencing (Supplementary Figure 1)
3. Segregation rates and growth of *mir2118* mutants (Supplementary Figures 2 and 4)
4. Developmental expression analysis of genes located in the miR2118-deleted regions (Supplementary Figure 3)
5. Developmental expression analysis of 21-nt phasiRNAs during anther development, from stage1 to stage 6 (Supplementary Figure 7).
6. Discussion of AGO1 and miR2118 functions including the analysis of proteome and the expression of AGO1b/d in *mir2118* mutants (Supplementary Table 5 and Supplementary Figure 9)
7. Separated tables in Figures according to Nature Communications Policy
8. 24nt-phasiRNAs analysis in *mir2118* mutants (Supplementary Figure 8)
9. The small RNA analysis in the pistil was removed, due to the lack of replicates

Point-by-point response to the referees' comments

Reviewer #1 (Remarks to the Author):

In this manuscript, Araki et al. report an analysis of a rice *mir2118* mutant generated using CRISPR-Cas9. The mutant appears to be a partial *mir2118* mutant, as at least 4 (known) copies of *miR2118* are not knocked out, and mature miRNAs derived from these loci are detected in anther and pistil. The mutant exhibits defects in both male and female reproductive development, causing male & female sterility. The developmental defects in anthers appear to be thicker tapetum and delayed death of middle layer. The authors also identified *mir2118*-triggered phasiRNAs in the somatic cells of anther and observed an enrichment of 5'-U in these phasiRNAs, distinct from the meiocyte-enriched phasiRNAs that are enriched for 5'-C. The analyses provide insights into the role of a subset of the *miR2118* family in male (& female) reproductive development. However, the study is lacking some of the key data in support of the conclusions.

Below are a number of specific comments/suggestions that may help improve the manuscript.

Major issues:

1. There is no quantitative data (e.g., segregation ratios) provided for the genetic analysis of the mutant alleles. As suggested, we added the numeric data for segregation ratios of the *mil-1*, *mil-2* and *mil-3* lines in Supplementary Fig. 2a. The segregation ratios in the offspring of heterozygous plants backcrossed once to WT 'Nipponbare' showed Mendelian inheritance ratio; *mil-1* (WT/H:-/-, 48:20, $\chi^2 = 0.71$ for 3:1), *mil-2* (WT/H:-/-, 41:9, $\chi^2 = 1.31$ for 3:1), *mil-3* (WT/H:-/-, 35:8, $\chi^2 = 0.94$ for 3:1). The homozygous plants showed significant sterility under SD conditions, compared with WT and heterozygous plants (Supplementary Fig. 2b).

2. The nature of the three mutant alleles is not clearly defined. For example, are the genomic sequences identical or different at the edited sites?

As commented, we further provided more detailed descriptions of the three mutant lines in the main text (Page 6; lines 5-26), and as figures in Figure 1 and Supplementary Figure 1. Sequencing analysis (whole genome sequence and sanger sequence) of the three lines demonstrated that the cleavage sites in these lines were identical (Figure 1b), while we identified hundreds of non-overlapping genetic variants in the lines

(Supplementary Figure 1e). Therefore, in the revision, we renamed the three lines as “*mir2118-1-1*, *mir2118-1-2*, and *mir2118-1-3* (*mi1-1*, *mi1-2*, and *mi1-3*)”.

3. The deletion of a 16-kb genomic region using two guide RNAs is remarkable. However, it seems that the deleted region includes at least an intact protein-coding gene and a portion of another protein-coding gene, but the authors did not rule out (or discuss) these genes as potential causal genes of the mutant phenotypes.

We agree with the reviewer regarding this point. Therefore, in the revised manuscript, we further analyzed putative genes located in the deleted region. According to the annotation in RAP-DB (<https://rapdb.dna.affrc.go.jp/>), there are six genes, Os04g0435300, Os04g0435325, Os04g0435350, Os04g0435402, Os04g0435450 and Os04g0435475, in the deleted regions, while the five genes except Os04g0435475/LOC_04g35550 were not annotated in MSU Rice Genome Annotation Project (<http://rice.plantbiology.msu.edu/>). Four genes (Os04g0435325, Os04g0435350, Os04g0435402 and Os04g0435450) are corresponding to the pre-miR2118 gene loci (Supplementary Figure 3a). Therefore, we focused on the remaining two hypothetical genes, Os04g0435300 and Os04g0435475/LOC_04g35550, and analyzed for the expression during rice development. Os04g0435300 and Os04g0435475/LOC_04g35550 were ubiquitously expressed in the rice tissues including leaves, while one of the *mir2118* precursor *pre-miR2118o* was specifically expressed at Stage 2 during anther development (Supplementary Figure 3b). Although we could not completely rule out the possibility that the loss of these genes have effects on the anther phenotypes, considering the ubiquitous expression patterns and the hypothetical nature of the genes, the reproductive defects in *mir2118* is most likely due to the loss of function of miR2118 by the deletion (Page7, lines11-18; Supplementary Figure3).

4. On a related note, the authors state that “the phenotypic variations of the two alleles might originate from the gRNA sequences used for the deletion of the loci that have high homology with the miR2118 recognition sites in over 700 reproductive lincRNAs”, implying editing at *21-PHAS* loci; if this is the case, whole-genome sequencing of the two mutants, which is done in this study, may identify mutations in the *21-PHAS* loci. However, this possibility is not discussed.

We thank the reviewer for the comment. We further analyzed genomic variations present in *mi1-1*, *mi1-2* and *mi1-3*, using the whole-genome sequencing data. Variant searches against the Nipponbare genome, however, did not detect genomic variants of deletion or/and insertion within *21PHAS* loci (body) in *mi1-1*, *mi1-2*, and

mi1-3, while six and two variants were detected within 1 kb upstream or downstream of *21PHAS* in *mi1-2*, and *mi1-3*, respectively (Supplementary Figure 1f). Accordingly, we revised the paragraph in the main text to incorporate the results (Page 7, line 19~).

5. Some important information is missing in the figures/figure legends. In all figures except Supplementary Figure 1, it's not clear whether the *mir2118* mutants are compared to their wild-type siblings or wild-type plants that are not their siblings.

We regret the unclear explanations about the control in the previous manuscript. Except for Supplementary Figure 2, wild-type Nipponbare was used as a control plant. A more detailed description of our control experiments can be found in the methods section (page17, lines3-5), and in figure legend (Sup Figure2).

6. There does not seem to be a clear set of hypotheses about the roles of AGO1b/AGO1d. The dysregulation of the AGO genes in the *mir2118* mutant may simply suggest that they are regulated, either direct or indirectly, by *miR2118* or the *miR2118*-triggered phasiRNAs, instead of being directly involved in *miR2118*/phasiRNA biogenesis as stated in the manuscript. The latter is in fact not well supported by the available data.

We agree with the reviewer, and in the revised manuscript, we deemphasized the descriptions about the involvement of the AGOs in *miR2118*/phasiRNA biogenesis. Understanding the functions of AGO1b/1d are great interests in the related research fields, and would be the subject of future studies. Actually, we are currently conducting projects to understand the role of the AGOs during the rice reproduction. In the discussion part of this revision, we further added the expression analysis of *AGO1b/d* in *mir2118* mutants (Sup Figure 9) and the following sentences to discuss the possibilities of the observed reduction of the AGO proteins in *mir2118* (Page14 line10-17): "There are two possibilities to explain the reduction of AGO1b, AGO1d, and MEL1 proteins in *mir2118* mutants: (1) the reduction of *miR2118* or phasiRNAs may cause the destabilization of AGO1b/AGO1d/MEL1 proteins in the absence of *miR2118*/phasiRNAs; or (2) defects of anther wall development in the *mir2118* may reduce the anther wall cell mass, which then reduces the relative amounts of AGO1 proteins. Reduced AGO1b/d proteins with no difference of the mRNA levels in *mir2118* mutants (Supplementary Fig. 9) may support the first hypothesis, in which the proteins are associated with the anther wall U-rich phasiRNAs or U-*miR2118* family members."

7. The writing of the manuscript needs polishing, perhaps by a native speaker. There are many issues in the

writing that make it taxing to follow. For example, on page 7, I don't agree that the results indicate that mir2118 causes defects in the elongation of OUTER anther locules, as the change in the shape of the outer locules is likely a consequence of curled inner locules – as described in the manuscript.

The revised manuscript was further edited by a professional editing service (Editage). We also revised the pointed sentences about the shape of anther locules in the main text (Page8 line14-16).

Minor issues:

8. Supplementary Figure 1: labels of the y-axes are missing, so it's not clear what type of data is shown.

We thank the reviewer for the comment. We added labels indicating the “fertility Rates (%)” to the Y-axes of the Supplementary Figure (now Supplementary Figure 2b).

9. Supplementary Figures 2: It's not clear what error bars in panels b & c represent and how many plants are compared. The difference in plant height of mi-2 vs. wild type is not support by any form of statistical test. In addition, images of whole plants are only shown for the mi-1 allele, but not for the mi-2 allele in which the dwarf phenotype is observed.

We thank the reviewer for the comments, and now we added the number of replicates, the error bar types, and p-values calculated by statistical tests in all figure legends. In terms of the plant height in Sup. Fig. 2 in the previous manuscript (now in Supplementary Figure 4), the experiment had done without the fertilizer, which did not result in the optimal growth of the plants. We repeated the experiment for this revision by growing the mutant plants with a fertilizer (HypoNex; <https://www.hyponex.co.jp/en/>) under SD and LD conditions. In the experiment, no significant difference in plant height between WT and *mi1-1* or *mi1-2* were observed (Supplementary Figure 4). Currently the cause of the developmental phenotype observed in the previous experiment is not clear, except the absence of the fertilizer. We now added a plant picture and data for plant height of *mi1-1* and *mi1-2* under LD and SD conditions (Supplementary Fig.4), and more detailed description in the main text.

We also described the replicates and error bar types in legends of Figures and Supplementary Figures.

10. Supplemental Figure 4d looks like an interesting set of data showing overall reduction in phasiRNA abundance, but it's not clear whether it's the 21-nt or the 24-nt phasiRNAs. In addition, the data are derived from two replicates as indicated in the legend, but it's not clear whether they are mean or sum of the replicates.

We thank the reviewer for the comments. Every two replicates of total 21-nt phasiRNAs demonstrate in Supplementary Figure 6c, though previous figure showed the sum of two replicates of 21-nt phasiRNAs. We added a description to the figure legends to clarify the points.

11. Page 5: The authors state that “MEL1 is the only AGO protein known to bind phasiRNAs during reproduction” – it would be better to clarify that this is only in rice, as there are known AGO proteins loading reproductive phasiRNAs in other plant species (e.g., maize).

We agree that this is an important point. We changed the sentence “MEL1 is the AGO protein known to bind to phasiRNAs during reproduction in rice” (Page5 line11-12).

Reviewer #2 (Remarks to the Author):

The manuscript describes the role of miR2118-dependent regulatory module in rice anther wall development by gene editing. The author integrated small RNA profiling, proteome, and 3D-histochemical analyses of miR2118 mutants to reveal the novel miR2118 functions and the biogenesis of miR2118-dependent U-phasiRNA/AGO1 subfamilies in the anther wall development. Overall, the experiments were well-designed and results were interesting. The data looked robust and well presented.

Comments/queries that need to be addressed:

1. Authors produced miR2118 mutants by genome editing and obtained three independent mutant alleles, mir2118-1, mir2118-2, and mir2118-3. Whether the authors conducted experiments such as southern blot to confirm these three mutants were actually independent mutant alleles? In Fig 1b, only mi-1 and mi-2 were presented, how about mi-3? Authors should add it to Fig 1b.

In response to the reviewer’s comments, we further provided more detailed descriptions of the three mutant lines in the main text (Page 6-7), and figures in Figure 1b and supplementary Figure 1. Instead of southern blotting, we performed sequencing analysis of the three lines (whole genome sequencing and sanger sequencing), which showed that the cleavage sites in these lines were identical (the information was added to Figure 1b; confirmation of the alleles by PCR in Supplementary figure 1d), while we also identified hundreds of non-overlapping genetic variants in the lines (Supplementary figure 1e). In the revision, we renamed the three lines as “*mir2118-1-1*, *mir2118-1-2*, and *mir2118-1-3* (*mi1-1*, *mi1-2*, and *mi1-3*)”. Furthermore, the

generic variants data detected in *21PHAS* clusters (Supplementary Figure 1f), the segregation ratios and the sterility (Supplementary Figure 2) of three lines were also provided.

2. In manuscript, authors described that there was 16,230bp deleted in *ir2118-1*, *mir2118-2*, and *mir2118-3* (*mi-1*, *mi-2*, and *mi-3*) by whole-genome sequencing. So what is the PAM sequences to achieve such large fragment deletion? And what about the vector information?

In the response to the comment, the PAM sequence of the guide RNA used for the genome-editing was shown in Figure 1a, and also described in the main text. *miR2118b* and *miR2118n* that are flanking the deletion sites have the identical sequence to the guide RNA except one C/U mismatch in the PAM sequence (Figure 1a), although the cleavages occurred downstream of the PAM sequences. We also added further information for the vector information in methods section (page17 line14-18) and added an additional reference: “Mikami M, Toki S, Endo M. Comparison of CRISPR/Cas9 expression constructs for efficient targeted mutagenesis in rice. *Plant Mol Biol* **88**, 561-572 (2015)”.

3. Authors said microRNA2118 (*miR2118*) conserved among plants, is expressed at reproductive stages, causing the production of secondary phased small interfering RNAs (*phasiRNAs*) and it plays an important role in rice anther wall development. Does *miR2118* has the similar function in other plants?

In monocots (Maize, Brachypodium), *miR2118* mainly targets long non-coding RNAs, which are specifically expressed at reproductive stages. In contrast, in dicot, coding genes targeted by *miR2118* include the NBS-LRR genes involved in pathogen defense and transcription factors. These results suggest that the target RNAs of *miR2118* are diverged between monocots (reproductive noncoding RNAs) and dicots (coding genes). In 2019, it is reported that tomato *miR2118* (dicot) is involved in the pathogen defense via 21-nt *phasiRNAs* production (Canto-Pastor et al., 2019). However, the molecular functions of *miR2118* during the reproductive stages are not fully understood in monocots (Page 4 line23-28).

In this study, we showed the function of *miR2118* in the rice reproduction. Numerous reproductive long non-coding RNAs, targeted by *miR2118*, have been reported in monocots other than rice, suggesting the importance of *miR2118* as a key regulator of the reproductive process in other species.

4. Authors found that there was no significant difference in plant height between WT and *mi-1*, while *mi-2* showed a dwarf phenotype under SD condition. And is there any difference between WT and mutants under

LD condition? Why did authors choose LD and SD conditions to investigate the role of miR2118 in rice anther wall development?

In terms of the plant height in Sup. Fig. 2 in the previous manuscript (now in Supplementary Figure 4), the experiment had done without the fertilizer, which did not result in the optimal growth of the plants. We repeated the experiment for this revision by growing the mutant plants with a fertilizer (HypoNex; <https://www.hyponex.co.jp/en/>) under SD and LD conditions. In the experiment, no significant difference in plant height between WT and *mi1-1* or *mi1-2* were observed (Supplementary Figure 4). Currently the cause of the developmental phenotype observed in the previous experiment is not clear, except the absence of the fertilizer. We now added a plant picture and data for plant height of *mi1-1* and *mi1-2* under LD and SD conditions (Supplementary figure 4), and more detailed description in the main text (Page7 line19-22).

It has been reported that the mutants of factors controlling the phasiRNA pathway in rice exhibit temperature-sensitivity or photoperiod-sensitivity (Song et al., 2012; Fan et al., 2016). Especially, *Long-day specific male fertility associated RNA (LDMAR)* and *PMSIT*, long non-coding RNA precursors for small RNAs, regulate photoperiodic male sterility, and the mutants show LD specific sterility in male (Ding et al., 2012, Fan et al., 2016). Therefore, we examined expressions of *PMSIT* and photoperiodic factor genes in Supplementary Figure 10. However, there was no significant difference in expression of those genes in *miR2118*, suggesting the sterility in *mir2118* is independent of the photoperiodic flowering pathway. We refer the reference of *PMSIT* in the beginning of first paragraph of results (Page6 line24-26), and discussed about this with expression analysis of photoperiodic genes in this revise (Page15 line25~).

5. Authors classified rice anther development into six stages and selected Stage 2 to analyze the overall structure of anthers. And moreover, 0.5 mm anthers at Stage 2 were also used to perform the transcriptome analysis of small RNAs. So how about the different/differential expressions of small RNAs of the other five stages anthers between WT and miR2118 mutants? I suggested authors supply these data on the manuscript.

In the response to the comment, we further examined expression patterns of representative phasiRNAs (phasiRNA324 and 798) in the WT and *mi1-1* anthers through stage1 to stage 6, and the results were added as Supplementary Figure 7. These phasiRNAs showed reductions throughout the six stages in the anther development in *mir2118*, suggesting that miR2118 is involved in the biogenesis of phasiRNAs during the anther development.

6. Expressions of encoding protein genes play important roles in rice anther development. Therefore, I wonder the differentially expressed genes between WT and miR2118 mutants. These data can be added to the manuscript if possible.

We agree with the reviewer. In the revision, we added the list of differentially expressed proteins (<0.8 fold change) in *mi1-1* mutant as Supplementary Table 5. The list includes basic helix-loop-helix transcription factors TDR1 (TAPETUM DEGENERATION RETARDATION1) and TIP2 (TDR INTERACTING PROTEIN2), required for the specification of the inner anther wall layers and tapetum differentiation (Fu et al., 2014). These results suggest that miR2118 may affect the production of the anther wall specific transcription factors via anther differentiation directly or indirectly. These descriptions were also added in the result section (Page12 line7-11).

7. What's the role of AGO proteins in rice anther wall development? It should be clarified in the manuscript.

We agree with the reviewer, and in the revised manuscript, we deemphasized the descriptions about the involvement of the AGOs in miR2118/phasiRNA biogenesis. Understanding the functions of AGO1b/1d are great interests in the related research fields, and would be the subject of future studies. Actually, we are currently conducting projects to understand the role of the AGOs during the rice reproduction. In the discussion part of this revision, we further added the expression analysis of *AGO1b/d* in *mir2118* mutants (Sup Figure 9) and the following sentences to discuss the possibilities of the observed reduction of the AGO proteins in *mir2118* (Page14 line10-17): "There are two possibilities to explain the reduction of AGO1b, AGO1d, and MEL1 proteins in *mir2118* mutants: (1) the reduction of miR2118 or phasiRNAs may cause the destabilization of AGO1b/AGO1d/MEL1 proteins in the absence of miR2118/phasiRNAs; or (2) defects of anther wall development in the *mir2118* may reduce the anther wall cell mass, which then reduces the relative amounts of AGO1 proteins. Reduced AGO1b/d proteins with no difference of the mRNA levels in *mir2118* mutants (Supplementary Fig. 9) may support the first hypothesis, in which the proteins are associated with the anther wall U-rich phasiRNAs or U-miR2118 family members."

8. In present study, the inner mechanism of miR2118 in rice anther wall development is not so adequate and further research is suggested to be done to clarify the specific function of miR2118 in rice wall development.

We agree with the reviewer's comment. The underlying miR2118-mediated molecular mechanisms in the rice anther wall development remain unclear. In this revision, one paragraph is added to discuss the possibilities of miR2118 function: (1) dual functions in the specific anther wall cell layers, (2) secondary effects of the abnormal differentiation process of outer layers during pre-meiosis. Furthermore, we refer the publications of OCL4 and MSP1, which regulate outer cell layer in anther, and these mutants show the similar phenotype of *mir2118*. We discussed an interaction between the soma and germ and/or a non-cell autonomous regulation via miR2118/secondary siRNAs (Page15 line4-18).

We are currently conducting research projects to understand the role of miR2118 and the AGOs during the rice reproduction by focusing on the AGO1b/1d and the interacting miR2118 and/or U-phaseRNAs, the results of which may be published hopefully in the near future.

REVIEWERS' COMMENTS:

Reviewer #1 (Remarks to the Author):

Overall, the authors adequately addressed the major concerns I raised for the first version of the manuscript. Key new data have been added, and the writing has been greatly improved. I am glad to see that the authors discussed the protein-coding genes that are knocked out along with the miR2118 cluster. However, in Supplementary Figure 3, it seems that Os04g0435300 does show a preferential expression in the 0.5-mm anthers than in other tissues/organs – rather than ubiquitously expressed, as the authors suggested. Moreover, a ubiquitously expressed gene may still have specialized functions in distinct tissues. I am not asking for more experiments to rule out the protein-coding genes that are knocked out in the current study, but the authors need to be objective about the data and explicitly discuss the limitation of the approach. Below are a few other (minor) comments:

1. It's unclear why the authors used a "-1" in the middle of the mutant names; it does not help with distinguishing the three alleles.
2. Supplemental Figure 6c: a statistical test should be included for the differences.
3. Supplemental Figure 10a: Many of the error bars overlap and therefore cause confusion in the interpretation of the data. It would be great if the authors could figure out a way to fix this. (My apologies that I did not point this out in the first round of review.)
4. I'm glad to see that the authors added discussion about PMS1T, the known 21-PHAS locus that has previously been shown to influence male fertility of rice; suggesting the miR2118 cluster analyzed in this work functions in an independent pathway than PMS1T. However, why didn't the author show any data for LDMAR, another male fertility-associated lncRNA locus that the authors mentioned in their response to the other reviewer?
5. Line 12: should the first gene ID be Os04g0435300 (see Supplementary Figure 3) instead of Os04g043500?

Reviewer #2 (Remarks to the Author):

Almost all the queries that need to be addressed in the revised manuscript entitled "miR2118-dependent U-rich phasiRNA production in rice anther wall development" have been modified carefully. The authors didn't perform experiments to confirm the inner mechanism of miR2118 in rice anther wall development but added much efficient text supplement to prove the idea.

Responses to reviewers' comments

We would like to thank the reviewers for their valuable comments on our revision. We have made changes in this revision of the manuscript in responses to the reviewer's comments as below.

Reviewer #1 (Remarks to the Author):

Overall, the authors adequately addressed the major concerns I raised for the first version of the manuscript. Key new data have been added, and the writing has been greatly improved. I am glad to see that the authors discussed the protein-coding genes that are knockout along with the miR2118 cluster. However, in Supplementary Figure 3, it seems that Os04g0435300 does show a preferential expression in the 0.5-mm anthers than in other tissues/organs – rather than ubiquitously expressed, as the authors suggested. Moreover, a ubiquitously expressed gene may still have specialized functions in distinct tissues. I am not asking for more experiments to rule out the protein-coding genes that are knocked out in the current study, but the authors need to be objective about the data and explicitly discuss the limitation of the approach. Below are a few other (minor) comments:

As commented, we could not completely rule out the possibility of these genes' effects on the anther phenotypes. Therefore, we removed the word “ubiquitously” and added the sentence in the *mir2118* mutant part of the result as follows (Page 7, line 18-20).

“~ while we could not exclude the possibilities that the loss of Os04g0435300 and Os04g0435475/LOC_04g35550 had effects on the observed phenotypes in *mir2118*.”

1. It's unclear why the authors used a “-1” in the middle of the mutant names; it does not help with distinguishing the three alleles.

As commented, we removed “-1” from the name of lines throughout the manuscript.

2. Supplemental Figure 6c: a statistical test should be included for the differences.

As suggested, we performed statistical tests on the data and *p*-values were indicated in the Supplementary Figure 6c.

3. Supplemental Figure 10a: Many of the error bars overlap and therefore cause confusion in the interpretation of the data. It would be great if the authors could figure out a way to fix this. (My apologies that I did not point this out in the first round of review.)

We thank the reviewer for the comment. We changed these graphs from lines to bar graphs (Supplementary Figure 11).

4. I'm glad to see that the authors added discussion about PMS1T, the known 21-PHAS locus that has previously been shown to influence male fertility of rice; suggesting the miR2118 cluster analyzed in this work functions in an independent pathway than PMS1T. However, why didn't the author show any data for LDMAR, another male fertility-associated lncRNA locus that the authors mentioned in their response to the other reviewer?

Although *LDMAR* is a long non-coding RNA precursor for “small RNAs” as commented, it remains unknown whether miR2118 targets *LDMAR* and 21-nt phased small RNAs are produced from it (Ding et al., 2012). Therefore, we focused on *PMS1T* for expression analysis in this study.

5. Line 12: should the first gene ID be Os04g0435300 (see Supplementary Figure 3) instead of

Os04g043500?

We thank the reviewer for the comment. We corrected the ID number (Page7 line13).

Reviewer #2 (Remarks to the Author):

Almost all the queries that need to be addressed in the revised manuscript entitled "miR2118-dependent U-rich phasiRNA production in rice anther wall development" have been modified carefully. The authors didn't perform experiments to confirm the inner mechanism of miR2118 in rice anther wall development but added much efficient text supplement to prove the idea.

We would like to thank the reviewer for their valuable comments on our revision. In our subsequent projects, we would like to clarify the molecular function of miR2118 in anther wall development, together with the identification of Argonaute proteins responsible for the process.